# Nickel(II) Coordination Polymers Supported by Bis-pyridyl-bis-amide and Angular Dicarboxylate Ligands: Role of Ligand Flexibility in Iodine Adsorption

**DOI:** 10.3390/ijms23073603

**Published:** 2022-03-25

**Authors:** Wei-Te Lee, Tsung-Te Liao, Jhy-Der Chen

**Affiliations:** Department of Chemistry, Chung Yuan Christian University, Chung Li 32023, Taiwan; lku23y230230@gmail.com (W.-T.L.); zxc0927808262@gmail.com (T.-T.L.)

**Keywords:** coordination polymer, crystal structure analysis, entanglement, iodine adsorption

## Abstract

Reactions of *N‚N’*-bis(3-pyridylmethyl)oxalamide (**L^1^**), *N‚N’*-bis(4-pyridylmethyl)oxalamide (**L^2^**), or *N,N’*-bis(3-pyridylmethyl)adipoamide) (**L^3^**) with angular dicarboxylic acids and Ni(II) salts under hydro(solvo)thermal conditions afforded a series of coordination polymers: {[Ni(**L^1^**)(OBA)(H_2_O)]·H_2_O}_n_ (H_2_OBA = 4,4-oxydibenzoic acid), **1**, {[Ni(**L^1^**)(SDA)(H_2_O)_2_]·H_2_O·CH_3_OH}_n_ (H_2_SDA = 4,4-sulfonyldibenzoic acid), **2**, {[Ni(**L^2^**)(OBA)]·C_2_H_5_OH}_n_, **3**, {[Ni(**L^2^**)(OBA)]·CH_3_OH}_n_, **4**, {[Ni_2_(**L^2^**)(SDA)_2_(H_2_O)_3_]·5H_2_O}_n_, **5**, {[Ni_2_(**L^2^**)(SDA)_2_(H_2_O)_3_]·H_2_O·2C_2_H_5_OH}_n_, **6**, {[Ni(**L^3^**)(OBA)(H_2_O)_2_]·2H_2_O}_n_, **7**, {[Ni(**L^3^**)(SDA)(H_2_O)_2_]·2H_2_O}_n_, **8**, and {[Ni(**L^3^**)_0_._5_(SDA)(H_2_O)_2_]·0.5C_2_H_5_OH}_n_, **9**, which have been structurally characterized by using single-crystal X-ray crystallography. Complex **1** exhibits an interdigitated 2D layer with the 2,4L2 topology and **2** is a 2D layer with the **sql** topology, while **3** and **4** are 3D frameworks resulting from polycatenated 2D nets with the **sql** topology and **5** and **6** are 2-fold interpenetrated 3D frameworks with the **dia** topology. Complexes **7** and **8** are 1D looped chains and **9** is a 2D layer with the 3,4L13 topology. The various structural types in **1**–**9** indicate that the structural diversity is subject to the flexibility and donor atom position of the neutral spacer ligands and the identity of the angular dicarboxylate ligands, while the role of the solvent is uncertain. The iodine adsorption of **1**–**9** was also investigated, demonstrating that that the flexibility of the spacer **L^1^**–**L^3^** ligands can be an important factor that governs the feasibility of the iodine adsorption. Moreover, complex **9** shows a better iodine adsorption and encapsulates 166.55 mg g^−1^ iodine in the vapor phase at 60 °C, which corresponded to 0.38 molecules of iodine per formula unit.

## 1. Introduction

Coordination polymers (CPs) have rapidly developed into an active area of chemical research, not only for their interesting structures and distinctive topologies, but also potential applications in diverse areas such as gas storage, drug delivery, catalysis, luminescence sensing, and ionic conductivity [1,2,3,4]. The coordinate covalent bonds that link the metal ions and the spacer ligands result in multi-dimensional networks, which are also subject to the nature of hydrogen bonding and other weak interactions. Entanglement is a widely known phenomenon in the crystal engineering of CPs, because the self-assembled structures usually try to fill the space as much as possible to increase the stability of the structure. Three main entanglements involving interpenetration, polycatenation, and self-catenation [5] have been shown, while interdigitation represents that part of an adjacent identical structure is interlaced into the space of the other without interlock. Although many CPs with interesting structures and properties have been reported, the control of their dimensionality and structural type is still a challenge and a great effort is necessary for the ongoing investigation.

Radioactive iodine such as ^129^I represents one of the most critical fission products in nuclear waste disposal, which has a long radioactive half-life and is harmful to human health. Therefore, these iodine species have to be captured and disposed of effectively [6,7,8]. However, limited adsorption ability, high cost, and environmental issues with the materials under investigation has created demand for developing alternative materials. On the other hand, CPs showing tailorable structures and possessing pores may facilitate iodine adsorption through noncovalent interactions involving iodine and various sorption sites such as amino functionalities [6,7,8].

We have been committed to the synthesis and structural characterization of bis-pyridyl-bis-amide (bpba) ligands containing pyridyl nitrogen atoms and amide groups linked by different—(CH_2_)_n_—backbones [9,10,11,12]. Modification of ligand flexibility has been found important in changing the structural diversity. In this report, the **L^1^**–**L^3^** ligands with different flexibility (Figure 1) were reacted with the metal salts, together with the angular dicarboxylic acid, to explore the structural diversity of nine Ni(II) CPs in mixed-ligand systems. The effect of flexibility and the donor atom position of the spacer ligands on the structural diversity and iodine adsorption were also investigated.

## 2. Results and Discussion

### 2.1. Crystal Structure of ***1***

The single-crystal X-ray diffraction analysis showed that **1** crystallizes in the triclinic space group *P*ī. The asymmetric unit comprises one Ni(II) cation, one **L^1^** ligand, one OBA^2−^ ligand, and one coordinated and one co-crystallized water molecule. Figure 2a shows the coordination environment of the Ni(II) metal center, which is six-coordinated by two nitrogen atoms from two **L^1^** ligands [Ni-N = 2.064(2) − 2.090(2) Å], three oxygen atoms from two OBA^2−^ ligands [Ni-O = 2.039(16) − 2.195(16) Å], and one oxygen atom from the coordination water [Ni-O = 2.100(17) Å]. The Ni(II) cations are linked by the OBA^2−^ and **L^1^** ligands to form a 2D layer. If the Ni(II) units are defined as 4-connected nodes and the OBA^2−^ as 2-connected nodes, the structure can be simplified as a 2,4-connected net with the 2,4L2 topology (Figure 2b), determined using ToposPro [13]. Figure 2c shows the interdigitation.

### 2.2. Structure of ***2***

The crystal structure of **2** was solved in the triclinic space group *P*ī, revealing that each asymmetric unit consists of one Ni(II) cation, one **L^1^** ligand, one SDA^2−^ ligand, two coordinated and one co-crystallized water molecule, and one MeOH molecule. Figure 3a shows the coordination environment of the Ni(II) metal center, which is six-coordinated by two nitrogen atoms from **L^1^** ligands [Ni-N = 2.104(19) − 2.108(17) Å], four oxygen atoms from two SDA^2−^ ligands [Ni-O = 2.017(14) − 2.049(14) Å], and two coordination water molecules [Ni-O = 2.084(18) − 2.101(17) Å]. The Ni(II) cations are linked by SDA^2−^ and **L^1^** ligands to form a 2D layer. Considering the Ni(II) units as 4-connected nodes, the structure can be simplified as a 4-connected net with the **sql** topology (Figure 3b).

### 2.3. Structures of ***3*** and ***4***

Structures of **3** and **4** are similar but different in the co-crystallized solvents. Crystals of **3** and **4** conform to the monoclinic space group *P*2_1_/*c* and each asymmetric unit contains one Ni(II) cation, one **L^2^** ligand, one OBA^2−^ ligand, and one co-crystallized solvent molecule (EtOH, **3**; MeOH, **4**). Figure 4a displays a representative coordination environment of the Ni(II) metal center, which is six-coordinated by two nitrogen atoms from two **L^2^** ligands [Ni-N = 2.057(2) and 2.064(2) Å, **3**; 2.050(16) and 2.057(16) Å, **4**] and four oxygen atoms from two OBA^2−^ ligands [Ni-O = 2.051(17) − 2.207(19) Å, **3**; 2.050(14) − 2.209(14) Å, **4**]. The Ni(II) cations are linked by OBA^2−^ and **L^2^** ligands to form a 2D layer. If the Ni(II) units are defined as 4-connected nodes, the structure can be simplified as a 4-connected net with the **sql** topology (Figure 4b), showing a 2D → 3D polycatenation (Figure 4c).

### 2.4. Structure of ***5*** and ***6***

The structures of complexes **5** and **6** are similar but different in the co-crystallized solvents. Single-crystal X-ray diffraction analysis showed that **5** and **6** crystallize in the triclinic space group *P*ī. Each asymmetric unit consists of two Ni(II) cations, one **L^2^** ligand, one SDA^2−^ ligand, and three coordinated water molecules (two terminal and one bridging), together with five co-crystallized water molecules in **5** and one co-crystallized water molecule and two EtOH molecules in **6**, respectively. Figure 5a shows a representative coordination environment of the two Ni(II) metal centers. While the Ni(1) atom is six-coordinated by two nitrogen atoms from two **L^2^** ligands [Ni-N = 2.087(2) and 2.120(2) Å, **5**; 2.080(1) and 2.113(19) Å, **6**], four oxygen atoms from three SDA^2−^ ligands [Ni-O = 2.083(17) − 2.084(17) Å, **5**; 2.020(16) − 2.066(16) Å, **6**], and the bridged water molecules [Ni-O = 2.123(17) Å, **5**; 2.106(15) Å, **6**], the Ni(2) is six-coordinated by six oxygen atoms from three SDA^2−^ ligands [Ni-O = 2.021(18) − 2.029(17) Å, **5**; 2.020(16) − 2.041(16) Å, **6**] and three coordination water molecules [Ni-O = 2.077(15) − 2.091(17) Å, **5**; 2.065(15) − 2.102(18) Å, **6**]. The Ni(II) cations are linked by SDA^2−^ and **L^2^** ligands to form a 3D framework. Considering the di-nuclear Ni(II) units as 4-connected nodes, their structures can be simplified as 4-connected nets with the **dia** topology (Figure 5b), showing the 2-fold interpenetration (Figure 5c).

### 2.5. Structure of ***7***

The single-crystal X-ray diffraction analysis showed that **7** crystallizes in the monoclinic space group *C*2/*c*, with each asymmetric unit consisting of one Ni(II) cation, one **L^3^** ligand, one OBA^2−^ ligand, and one coordinated and two co-crystallized water molecules. Figure 6a shows the coordination environment of the Ni(II) metal center, which is six-coordinated by two nitrogen atoms from two **L^3^** ligands [Ni-N = 2.100(1) Å], four oxygen atoms from two OBA^2−^ ligands [Ni-O = 2.068(2) − 2.081(4) Å], and two coordinated water molecules [Ni-O = 2.068(2) − 2.081(4) Å]. The Ni(II) cations are linked by OBA^2−^ and **L^3^** ligands to form a 1D looped chain (Figure 6b).

### 2.6. Structure of ***8***

Crystals of **8** conform to the monoclinic space group *C*2/*c*, with each asymmetric unit consisting of one Ni(II) cation, one **L^3^** ligand, one SDA^2−^ ligand, and two coordinated and two co-crystallized water molecules. Figure 7a shows the coordination environment of the Ni(II) metal center, which is six-coordinated by two nitrogen atoms from two **L^3^** ligands [Ni-N = 2.100(1) Å], four oxygen atoms from two SDA^2−^ ligands [Ni-O = 2.068(2) − 2.081(4) Å], and two coordinated water molecules [Ni-O = 2.068(2) − 2.081(4) Å]. The Ni(II) cations are linked by the SDA^2−^ and the **L^3^** ligands to form a 1D looped chain (Figure 7b).

### 2.7. Structure of ***9***

The single-crystal X-ray diffraction analysis showed that **9** crystallizes in the triclinic space group *P*ī. The asymmetric unit consists of one Ni(II) cation, a half **L^3^** ligand, one SDA^2−^ ligand, two coordinated water molecules, and a half co-crystallized EtOH molecule. Figure 8a shows the coordination environment of the Ni(II) metal center, which is six-coordinated by one nitrogen atom from one **L^3^** ligand [Ni-N = 2.098(2) Å] and five oxygen atoms from two SDA^2−^ ligands [Ni-O = 2.036(17) − 2.169(17) Å], one **L^3^** ligand, and two coordinated water molecules [Ni-O = 2.068(2) − 2.081(4) Å]. The Ni(II) cations are linked by SDA^2−^ and **L^3^** ligands to form a 2D layer. If the Ni(II) cations are defined as 3-connected nodes and the **L^3^** ligand as 4-connected nodes, the structure of **9** can be simplified as a 3,4-connected net with the 3,4L13 topology (Figure 8b).

### 2.8. Ligand Conformations and Coordination Modes

By calculating the torsion angles of the long carbon chain and evaluating the orientations of the pyridyl rings and carbonyl group, the conformations of the bpba ligands can be expressed as follows [14]: When the torsion angle is 0 ≤ θ ≤ 90°, it is defined as gauche (G), and 90 < θ ≤ 180° as anti (A). Additionally, *cis* and *trans* conformations can also be shown if the two C=O groups are in the same and the opposite direction, respectively. Due to the different orientations adopted by the pyridyl nitrogen atoms and the amide oxygen atoms, three more conformations, *syn*-*syn*, *syn*-*anti*, and *anti*-*anti*, can also be found for bpba. Table 1 lists the ligand conformations and coordination modes of the organic ligands in complexes **1**–**9**. Noticeably, the **L^1^**, **L^2^**, and **L^3^** ligands in **1**–**8** bridge two metal ions through two pyridyl nitrogen atoms, while the **L^3^** ligand of **9** bridges four metal ions through two pyridyl nitrogen atoms and two amide oxygen atoms. The bpba ligands that bridge four metal ions are rare and can be found for the bis(N-pyrid-3-ylmethyl)suberoamide (L) in {[Cd(L)(1,4-NDC)]·2H_2_O}_n_ (1,4-H_2_NDC = naphthalene-1,4-dicarboxylic acid) and {[Cd_2_(L)(1,4-NDC)_2_]·3H_2_O}_n_ [15].

The OBA^2−^ and SDA^2−^ ligands show variable coordination modes. In **1**, the OBA^2−^ ligand bridges two Ni(II) ions with one chelation mode, while the dicarboxylate ligands in **2** and **7**–**9** bridge two Ni(II) ions, and each of the two carboxylate groups coordinates one Ni(II) ion through one oxygen atom. In **3** and **4**, the OBA^2−^ ligands bridge two Ni(II) ions with two chelation modes, while in **5** and **6**, the SDA^2−^ ligands bridge three Ni(II) ions, leaving one of the carboxylate oxygen atoms uncoordinated. Structural comparisons show that for the complexes with **L^1^** (**1** and **2**) or **L^2^** (**3**–**6**) ligands, the structural diversity is subject to the change of the angular dicarboxylate ligand, while the solvent is not influential. However, the structural diversity of those containing **L^3^** (**7**–**9**) is subject to the changes of both the angular dicarboxylate ligand and solvent. The different structural types between **8** and **9** demonstrate that the ligand conformation of the flexible **L^3^** is subject to the change of the solvent identity, resulting in AAA-*trans*-*syn*-*syn* and GAG-*trans*-*syn*-*anti* and leading to the formation of a 1D looped chain and 2D layer, respectively.

### 2.9. PXRD Patterns and Thermal Analysis

The experimental PXRD patterns of complexes **1**–**9** (Appendix A) match well with their corresponding simulated ones, which demonstrates that the purities of the bulk samples are good enough for further use. On the other hand, thermal gravimetric analysis (TGA) was carried out to examine the thermal decomposition of complexes **1**–**9**. The TGA curves are shown in Appendix A, and Table 2 lists the decomposition temperatures, showing two-step decomposition and indicating that the decomposition temperatures for organic ligands of the frameworks of **3** and **4** with the polycatenated frameworks are much higher among **1**–**9**.

### 2.10. Iodine Adsorption

Complexes **7**–**9** provide a unique opportunity to investigate the degree of iodine adsorption of bpba-based Ni(II) CPs, which have been executed at 25 and 60 °C and with time intervals of 30, 60, 120, 180, and 360 min, respectively. For each experiment, 0.05 mmol of the complex was placed in a 5 mL sample bottle inside a 50 mL one containing 100 mg of iodine, which was sealed, kept in the oven, and heated. The I_2_-adsorbed complex was then weighted and the amount of adsorbed I_2_ was calculated. Each experiment was repeated three times and the results were averaged (Appendix A). At 25 °C, the color of complex **7** changed from blue to green, while the color changed from blue to yellow at 60 °C (Appendix A). Complex **8** showed no color change at 25 and 60 °C (Appendix A), while the color of complex **9** changed from green to dark brown at these two temperatures (Appendix A).

Figure 9, Figure 10 and Figure 11 display the average weight changes per gram of **7**–**9** upon iodine adsorption at different time intervals and at 25 and 60 °C, respectively, while Table 3 summarizes the results, showing that the adsorption capacities of **7** and **8** are much poorer than **9**. With the increase of temperature from 25 to 60 °C, the absorption rate of iodine also showed a good increase for each complex. The different iodine adsorption capacities can be ascribed to their different diversity, which are 1D chains for **7** and **8** and a 2D layer for **9**, respectively, showing a maximum adsorption factor of 166.55 mg g^−1^ for complex **9** at 60 °C for 360 min, corresponding to 0.38 iodine molecules per unit cell. Powder X-ray diffraction (PXRD) patterns of the I_2_-adsorbed complexes **7**–**9** have been measured to confirm their stability. As shown in Appendix A, most of the experimental patterns are consistent with the theoretical ones, indicating that these iodine-adsorbed complexes remain stable. Only the PXRD patterns of **8** at 60 °C showed some changes after 180 min.

Energy dispersive X-ray (EDX) analysis of complexes **7**–**9** was performed after iodine adsorption (Appendix A), confirming the iodine uptake of **7**–**9**. The iodine-adsorbed samples of **7**–**9** can be regarded as the mixtures of iodine and their corresponding complexes. As expected, the amounts of iodine elements were different at three different spots of the samples selected for measurement, indicating the inhomogeneous distribution of iodine in the iodine-adsorbed samples. Table 4 shows the average weights and atomic percentages of the selected elements for the iodine-adsorbed **7**–**9**, confirming that **9** displays the best iodine adsorption capacity. In addition, experiments have also been performed for complexes **1**–**6** to evaluate their iodine adsorption capacities. As shown in Appendix A, the colors of **1**–**6** remained unchanged at both 25 and 60 °C, indicating minor or no iodine absorption, supported by their EDX data that showed no detectable iodine element (Appendix A).

The ability of the CPs to accommodate iodine molecules in the interchain/interlayer space perhaps governs the iodine adsorption capacity [16,17,18]. The solvent accessible volumes calculated by using the PLATON program [19] for **7**–**9** were 6.9, 9.3 and 13.7%, respectively, of the unit cell volume, indicating that the 2D **9** may accommodate more iodine than the 1D **7** and **8**. However, as shown in Appendix A, the solvent accessible volumes of complexes **1**–**6** were comparable to those of **7**–**9**, but these CPs revealed no iodine adsorption capacity, demonstrating the important role of the flexibility of the neutral spacer ligands, **L^1^**, **L^2^**, and **L^3^**, in determining the iodine adsorption capacities of **1**–**9**. The more flexible **L^3^** ligand that resulted in the 2D layer **9** with the 3,4L13 topology may be more susceptible to the changes of the ligand conformation upon the attack of the iodine molecules and thus better to accommodate the iodine molecules. On the other hand, the entangled complexes **1**–**6** comprising the rigid **L^1^** and **L^2^** ligands are not vulnerable to the changes of the frameworks upon the iodine attack, allowing undetectable iodine adsorption.

### 2.11. Gas Adsorption

Low-pressure N_2_ adsorption and desorption measurements were performed at 77 K for complexes **7**–**9**, and their isotherms are shown in Appendix A, showing a Brunauer–Emmet–Teller (BET) surface area and Langmuir surface of 10.00 and 15.22 m^2^/g for **7**, 9.04 and 14.47 m^2^/g for **8**, and 10.00 and 15.88 m^2^/g for **9**, respectively, indicating similar BET surface areas and Langmuir surfaces.

## 3. Experimental Section

### 3.1. General Procedures

Elemental analyses of (C, H, N) were performed on a PE 2400 series II CHNS/O (PerkinElmer Instruments, Shelton, CT, USA) or an Elementar Vario EL-III analyzer (Elementar Analysensysteme GmbH, Hanau, Germany). Infrared spectra were obtained from a JASCO FT/IR-460 plus spectrometer with pressed KBr pellets (JASCO, Easton, MD, USA). Thermal gravimetric analyses (TGA) were carried out on a TG/DTA 6200 over the temperature range of 30 to 900 °C at a heating rate of 10 °C min^−1^ under N_2_ (SEIKO Instruments Inc., Chiba, Japan). Powder X-ray diffraction patterns were carried out with a Bruker D8-Focus Bragg-Brentano X-ray powder diffractometer equipped with a CuKα (λ_α_ = 1.54178 Å) sealed tube (Bruker Corporation, Karlsruhe, Germany). Gas sorption measurements were conducted using a Micromeritics ASAP 2020 system (Micromeritics Instruments Co., Norcross, GA, USA).

### 3.2. Materials

The reagent Ni(OAc)_2_·4H_2_O was purchased from Alfa Aesar (Ward Hill, MA, USA), and 4,4′-sulfonyldibenzoic acid (H_2_SDA) and 4,4-oxydibenzoic acid (H_2_OBA) from Aldrich Chemical Co. (St. Louis, MO, USA). The ligands *N,N’*-bis(3-pyridylmethyl)oxalamide (**L^1^**), *N,N’*-bis(4-pyridylmethyl)oxalamide (**L^2^**), and *N,N’*-bis(3-pyridylmethyl)adipoamide (**L^3^**) were prepared according to published procedures, with some modifications [14].

### 3.3. Preparations

#### 3.3.1. {[Ni(**L^1^**)(OBA)(H_2_O)]·H_2_O}_n_, **1**

A mixture of Ni(CH_3_COO)_2_·2H_2_O (0.050 g, 0.20 mmol), **L^1^** (0.027 g, 0.10 mmol), and H_2_OBA (0.052 g, 0.20 mmol) in 10 mL of H_2_O was sealed in a 23 mL Teflon-lined steel autoclave, which was heated under autogenous pressure to 100 °C for two days, and then allowed to cool down gradually to room temperature for two days. Green crystals suitable for single-crystal X-ray diffraction were obtained. Yield: 0.023 g (32 %). Anal. Calcd for C_28_H_26_N_4_NiO_9_ (MW = 621.24): C, 54.13; N, 9.02; H, 4.22 %. Found: C, 53.92; N, 8.97; H, 3.88 %. FT-IR (cm^−1^): 3567(m), 3372(m), 3224(w), 3056(w), 2937(w), 1676(s), 1594(s), 1509(s), 1423(s), 1392(s), 1230(s), 1164(m), 1102(w), 1008(w), 1005(w), 865(w), 779(w), 705(m), 637(w).

#### 3.3.2. {[Ni(**L^1^**)(SDA)(H_2_O)_2_]·H_2_O·CH_3_OH}_n_, **2**

Complex **2** was prepared by following similar procedures for **1** except that Ni(CH_3_COO)_2_·2H_2_O (0.050 g, 0.20 mmol), **L^1^** (0.027 g, 0.10 mmol), and H_2_SDA (0.061 g, 0.20 mmol) in 8 mL of H_2_O and in 2 mL of MeOH were used. Green crystals were obtained. Yield: 0.035 g (49 %). Anal. Calcd for C_29_H_32_N_4_NiO_12_S (MW = 719.35): C, 48.42; N, 7.79; H, 4.48 %. Found: C, 48.08; N, 7.91; H, 4.34 %. FT-IR (cm^−1^): 3496(m), 3326(m), 3062(w), 2938(w), 1671(s), 1597(s), 1549(s), 1527(s), 1521(s), 1390(s), 1291(m), 1224(m), 1161(m), 1106(m), 954(w), 864(w), 794(w), 729(s), 617(m).

#### 3.3.3. {[Ni(**L^2^**)(OBA)]·C_2_H_5_OH}_n_, **3**

Complex **3** was prepared by following similar procedures for **1** except that Ni(CH_3_COO)_2_·2H_2_O (0.050 g, 0.20 mmol), **L^2^** (0.027 g, 0.10 mmol), and H_2_OBA (0.052 g, 0.20 mmol) in 8 mL of H_2_O and in 2 mL of EtOH were used. Green crystals were obtained. Yield: 0.036 g (57 %). Anal. Calcd for C_30_H_28_N_4_NiO_8_ (MW = 631.27): C, 57.08; N, 8.88; H, 4.47 %. Found: C, 56.67; N, 8.54; H, 3.79 %. FT-IR (cm^−1^): 3454(s), 2370(w), 2305(w), 1671(m), 1623(m), 1515(w), 1503(w), 1418(m), 1225(m), 1153(w), 1065(w), 866(w), 799(w), 648(w).

#### 3.3.4. {[Ni(**L^2^**)(OBA)]·CH_3_OH }_n_, **4**

Complex **4** was prepared by following similar procedures for **1** except that Ni(CH_3_COO)_2_·2H_2_O (0.050 g, 0.20 mmol), **L^2^** (0.027 g, 0.10 mmol), and H_2_OBA (0.052 g, 0.20 mmol) in 8 mL of H_2_O and in 2 mL of MeOH were used. Green crystals were obtained. Yield: 0.025 g (41 %). Anal. Calcd for C_29_H_26_N_4_NiO_8_ (MW = 617.25): C, 56.43; N, 9.08; H, 4.25 %. Found: C, 56.15; N, 8.79; H, 4.51 %. FT-IR (cm^−1^): 3449(s), 3368(s), 3062(w), 2923(w), 2371(w), 1679(s), 1595(s), 1534(s), 1499(s), 1422(s), 1223(s), 1159(s), 1016(w), 873(m), 776(m), 659(m), 653(m).

#### 3.3.5. {[Ni_2_(**L^2^**)(SDA)_2_(H_2_O)_3_]·5H_2_O}_n_, **5**

Complex **5** was prepared by following similar procedures for **1** except that Ni(CH_3_COO)_2_·2H_2_O (0.050 g, 0.20 mmol), **L^2^** (0.027 g, 0.10 mmol), and H_2_SDA (0.061 g, 0.20 mmol) in 10 mL of H_2_O were used. Green crystals were obtained. Yield: 0.026 g (28 %). Anal. Calcd for C_42_H_46_N_4_Ni_2_O_22_S_2_ (MW = 1140.37): C, 44.24; N, 4.91; H, 4.07 %. Found: C, 43.93; N, 4.73; H, 4.03 %. FT-IR (cm^−1^): 3440(s), 2369(w), 2318(w), 1638(m), 1516(w), 1401(w), 1165(w), 1045(w), 890(w), 745(w), 697(w).

#### 3.3.6. {[Ni_2_(**L^2^**)(SDA)_2_(H_2_O)_3_]·H_2_O·_2_C_2_H_5_OH}_n_, **6**

Complex **6** was prepared by following similar procedures for **1** except that Ni(CH_3_COO)_2_·2H_2_O (0.050 g, 0.20 mmol), **L^2^** (0.027 g, 0.10 mmol), and H_2_SDA (0.061 g, 0.20 mmol) in 8 mL of H2O and in 2 mL of EtOH were used. Green crystals were obtained. Yield: 0.035 g (30 %). Anal. Calcd for C_46_H_50_N_4_Ni_2_O_20_S_2_ (MW = 1160.44): C, 47.61; N, 4.83; H, 4.34 %. Found: C, 47.96; N, 4.81; H, 4.01 %. FT-IR (cm^−1^): 3709(w), 3636(w), 3478(m), 3392(m), 3231(w), 3058(w), 2923(w), 2054(w), 1680(m), 1636(m), 1558(m), 1509(m), 1400(s), 1284(m), 1159(m), 1097(m), 1012(m), 835(m), 738(s), 617(m).

#### 3.3.7. {[Ni(**L^3^**)(OBA)(H_2_O)_2_]·2H_2_O}_n_, **7**

Complex **7** was prepared by following similar procedures for **1** except that Ni(CH_3_COO)_2_·2H_2_O (0.050 g, 0.20 mmol), **L****^3^** (0.033 g, 0.10 mmol), and H_2_OBA (0.052 g, 0.20 mmol) in 8 mL of H_2_O and 2 mL of MeOH were used. Green crystals were obtained. Yield: 0.048 g (67 %). Anal. Calcd for C_32_H_38_N_4_NiO_11_ (MW = 713.37): C, 53.88; N, 7.85; H, 5.37 %. Found: C, 53.32; N, 7.73; H, 5.28 %. FT-IR (cm^−1^): 3502(m), 3250(w), 3076(w), 2923(w), 2372(w), 2305(w), 1636(m), 1600(m), 1541(m), 1384(m), 1227(m), 1156(w), 1026(w), 726(w).

#### 3.3.8. {[Ni(**L^3^**)(SDA)(H_2_O)_2_]·2H_2_O}_n_, **8**

Complex **8** was prepared by following similar procedures for **1** except that Ni(CH_3_COO)_2_·2H_2_O (0.050 g, 0.20 mmol), **L^3^** (0.033 g, 0.10 mmol), and H_2_SDA (0.061 g, 0.20 mmol) in 10 mL of H_2_O were used. Green crystals were obtained. Yield: 0.035 g (49 %). Anal. Calcd for C_32_H_38_N_4_NiO_12_S (MW = 761.43): C, 50.48; N, 7.36; H, 5.03 %. Found: C, 50.49; N, 7.04; H, 5.05 %. FT-IR (cm^−1^): 3530(m), 3488(m), 3253(m), 3068(m), 2931(m), 1637(s), 1554(s), 1384(s), 1295(w), 1158(m), 1101(m), 1014(m), 835(m), 789(m), 745(m), 701(m), 620(m), 557(m).

#### 3.3.9. {[Ni(**L^3^**)_0_._5_(SDA)(H_2_O)_2_]·0.5C_2_H_5_OH}_n_, **9**

Complex **9** was prepared by following similar procedures for **1** except that Ni(CH_3_COO)_2_·2H_2_O (0.050 g, 0.20 mmol), **L^3^** (0.033 g, 0.10 mmol), and H_2_SDA (0.061 g, 0.20 mmol) in 10 mL of H_2_O were used. Green crystals were obtained. Yield: 0.038 g (65 %). Anal. Calcd for C_24_H_26_N_2_NiO_9_._5_S (MW = 585.24): C, 49.06; N, 4.79; H, 4.48 %. Found: C, 49.22; N, 4.58; H, 4.47 %. FT-IR (cm^−1^): 3450(m), 3298(m), 2929(m), 2225(w), 1932(w), 1626(s), 1557(s), 1395(w), 1279(m), 1155(m), 1009(m), 1002(m), 822(m), 736(m), 616(m), 563(m).

### 3.4. X-ray Crystallography

The phase purities of complexes **1**–**9** were verified by using powder X-ray diffraction (PXRD). As shown in Appendix A, the experimental PXRD patterns match well with the corresponding simulated ones, indicating that the bulk samples are pure enough.

Single-crystal X-ray diffraction data for complexes **1**–**9** were collected on a Bruker AXS SMART APEX II CCD diffractometer with a graphite-monochromated MoKα (λ_α_ = 0.71073 Å) radiation at 296 K [20]. Data reduction and absorption correction were performed by using standard methods with well-established computational procedures [21]. Some of the heavier atoms were located by the direct or Patterson method and the remaining atoms were found in a series of different Fourier maps and least-square refinements, while the hydrogen atoms were added by using the HADD command in SHELXTL. Basic information pertaining to crystal parameters and structure refinement is listed in Table 5.

## 4. Conclusions

Nine Ni(II) coordination polymers containing angular dicarboxylate ligands and bpba ligands with different flexibility and donor atom positions have been synthesized under hydro(solvo)thermal conditions. The various structural types in **1**–**9** showed that the structural diversity of these Ni(II) CPs is subject to the flexibility and donor atom position of the bpba ligands and the identity of the angular dicarboxylate ligands. The **L^3^** ligand of **9** adopted a rare coordination mode that bridges four metal ions through two pyridyl nitrogen atoms and two amide oxygen atoms. Complex **9** showed an iodine adsorption factor of 166.55 mg g^−1^ at 60 °C for 360 min, while that of **7** and **8** was low and **1**–**6** showed undetectable iodine adsorption, demonstrating that the flexibility of the bpba ligands is important in governing the iodine adsorption of the bpba-CPs supported by the angular dicarboxylate ligands.

## Figures and Tables

**Figure 1 ijms-23-03603-f001:**
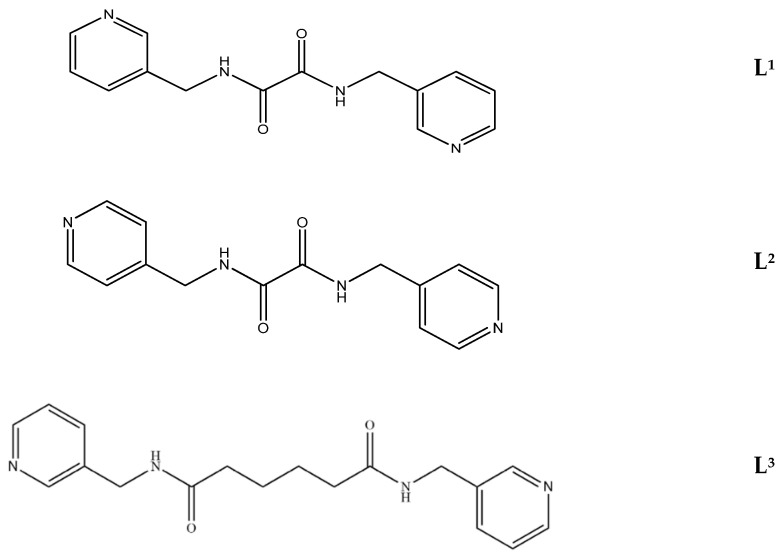
Structures of the ligands **L^1^**–**L^3^**.

**Figure 2 ijms-23-03603-f002:**
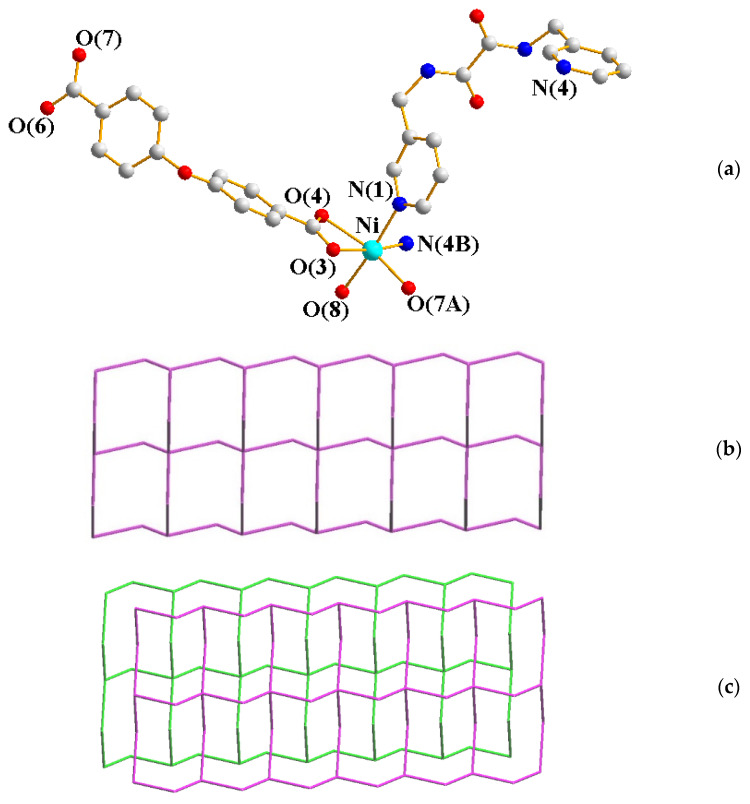
(**a**) Coordination environment of Ni(II) ion in **1**. Symmetry transformations used to generate equivalent atoms: (A) x, y + 1, z − 1; (B) x − 1, y, z. (**b**) A drawing showing the 2,4L2 topology. (**c**) A drawing showing the interdigitation.

**Figure 3 ijms-23-03603-f003:**
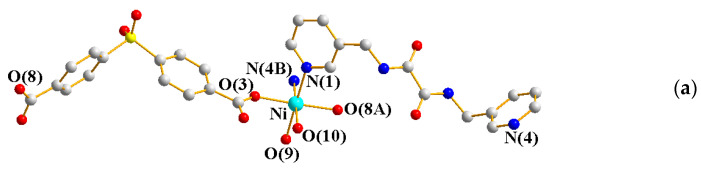
(**a**) Coordination environment of Ni(II) ion in **2**. Symmetry transformations used to generate equivalent atoms: (A) x + 1, y, z + 1; (B) x, y, z − 1. (**b**) A drawing showing the **sql** topology.

**Figure 4 ijms-23-03603-f004:**
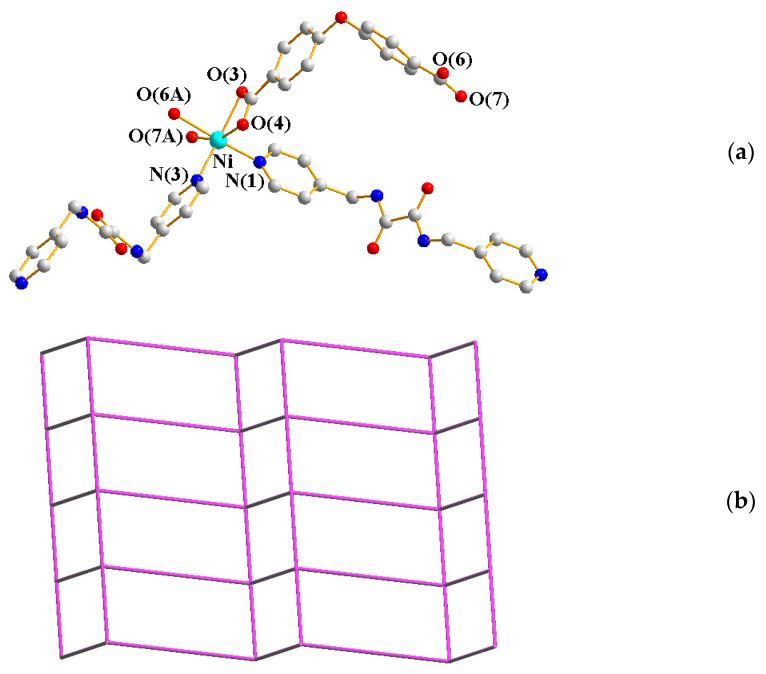
(**a**) Coordination environment of the Ni(II) ion in **3** and **4**. Symmetry transformations used to generate equivalent atoms: (A) x, y + 1, z. (**b**) A drawing showing the **sql** topology. (**c**) A drawing showing the polycatenation.

**Figure 5 ijms-23-03603-f005:**
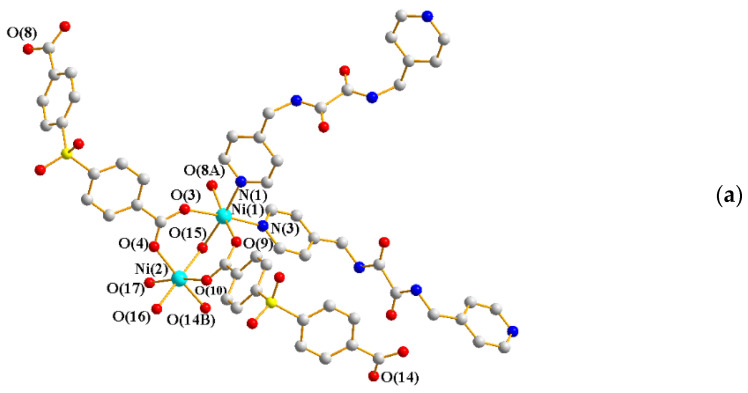
(**a**) Coordination environment of Ni(II) ion in **5** and **6**. Symmetry transformations used to generate equivalent atoms: (A) −x + 2, −y + 2, −z − 1; (B) −x + 1, −y + 1, -z for **5** and (A) −x + 1, −y, −z + 1; (B) −x + 2, −y + 1, −z for **6**. (**b**) A drawing showing the **dia** topology. (**c**) A drawing showing the 2-fold interpenetration.

**Figure 6 ijms-23-03603-f006:**
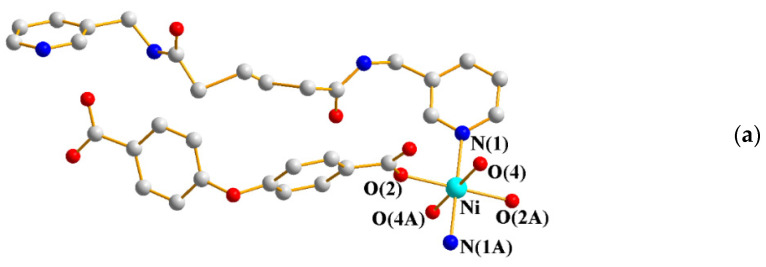
(**a**) Coordination environment of Ni(II) ion in **7**. Symmetry transformations used to generate equivalent atoms: (A) x + 1/2, −y + 3/2, −z. (**b**) A drawing showing the 1D structure.

**Figure 7 ijms-23-03603-f007:**
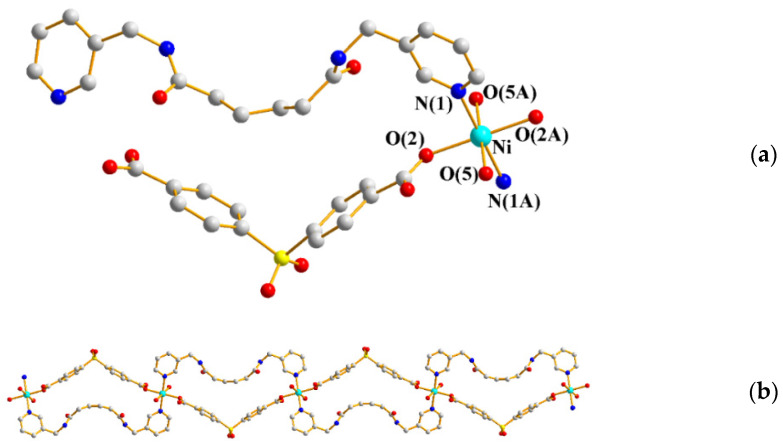
(**a**) Coordination environment of Ni(II) ion in **8**. Symmetry transformations used to generate equivalent atoms: (A) −x + 1/2, −y + 1/2, −z; (B) −x, y, −z + 1/2. (**b**) A drawing showing the 1D structure.

**Figure 8 ijms-23-03603-f008:**
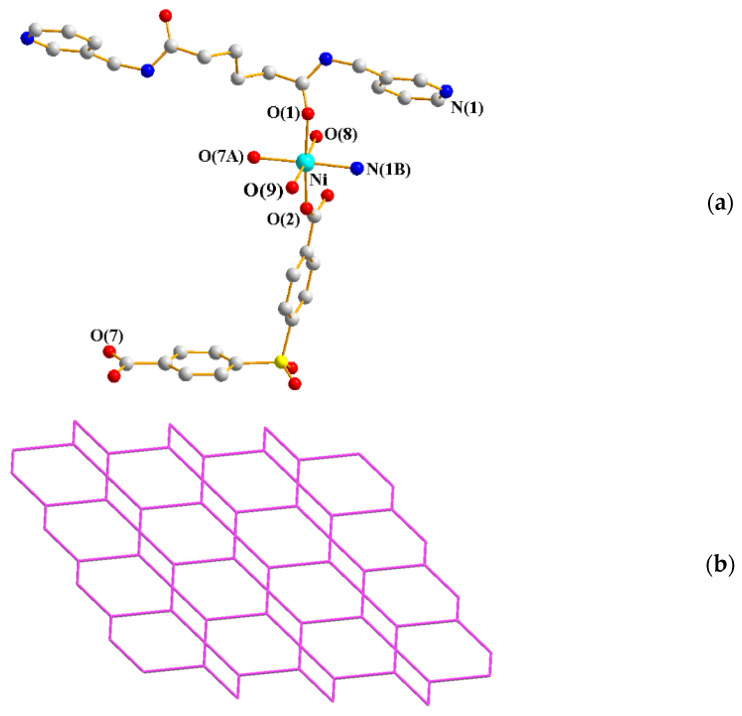
(**a**) Coordination environment of Ni(II) ion in **9**. Symmetry transformations used to generate equivalent atoms: (A) −x + 1, −y + 2, −z + 2; (B) −x + 1, −y + 1, −z + 1. (**b**) A drawing showing the 3,4L13 topology.

**Figure 9 ijms-23-03603-f009:**
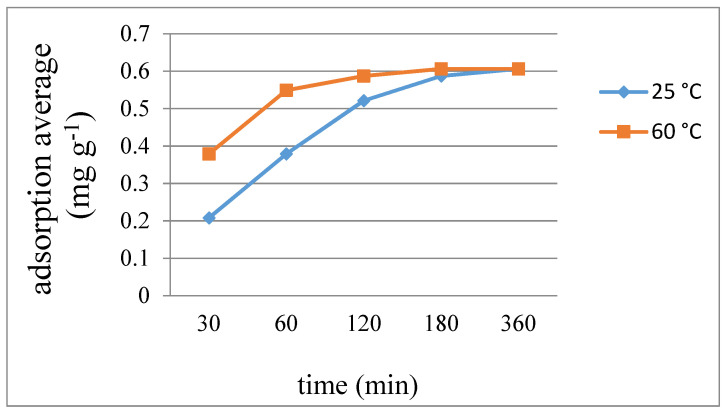
Average weight changes of complex **7** upon iodine adsorption.

**Figure 10 ijms-23-03603-f010:**
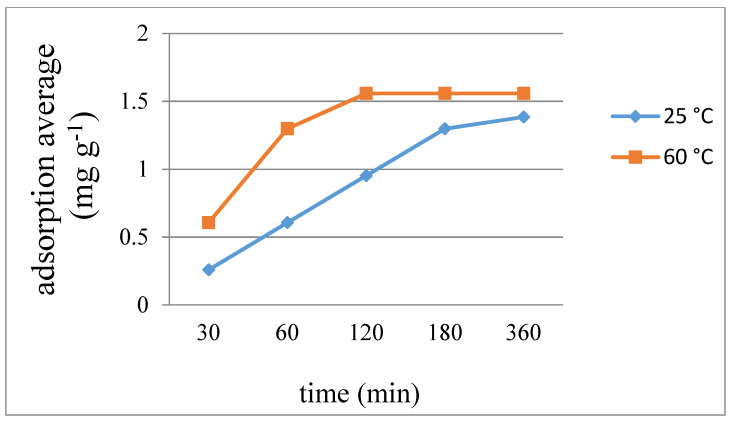
Average weight changes of complex **8** upon iodine adsorption.

**Figure 11 ijms-23-03603-f011:**
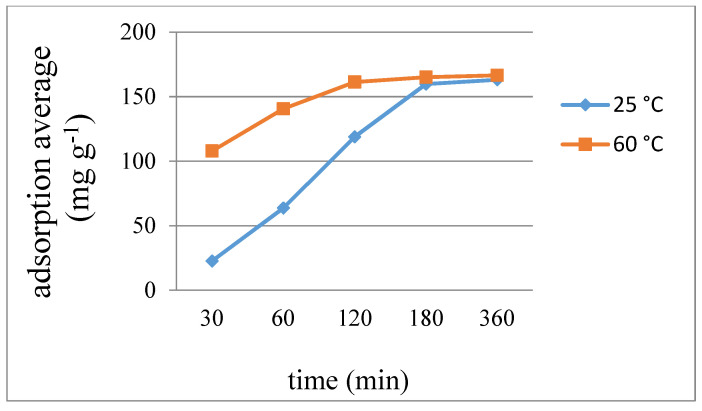
Average weight changes of complex **9** upon iodine adsorption.

**Table 1 ijms-23-03603-t001:** Ligand conformations and bonding modes of **1**–**9**.

Complexes	Conformation	Coordination Mode
**1**	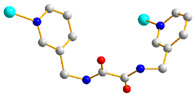 *trans*-*anti*-*anti*	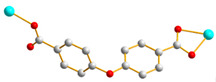 *μ_2_*-*κ*O*κ*O’:*κ*O’’
**2**	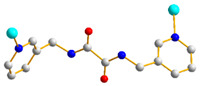 *trans*-*syn*-*anti*	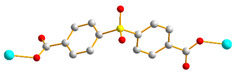 *μ_2_*-*κ*O:*κ*O’
**3**, **4**	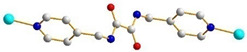 *trans*	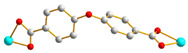 *μ_2_*-*κ*O*κ*O’:*κ*O’’*κ*O’’’
**5**, **6**	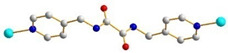 *trans*	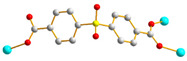 *μ_3_*-*κ*O:*κ*O’:*κ*O’’
**7**	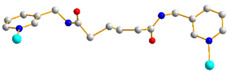 AAA-*trans*-*syn*-*syn*	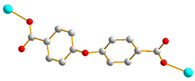 *μ_2_*-*κ*O:*κ*O’
**8**	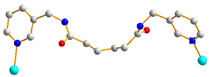 AAA-*trans*-*syn*-*syn*	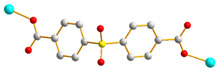 *μ_2_*-*κ*O:*κ*O’
**9**	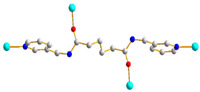 GAG-*trans*-*syn*-*anti*	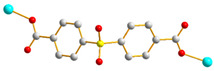 *μ_2_*-*κ*O:*κ*O’

**Table 2 ijms-23-03603-t002:** Thermal properties of complexes **1**–**9**.

Complex	Weight Loss of Solvent °C (Calc/Found), %	Weight Loss of Ligand °C (Calc/Found), %
**1**	2 H_2_O	**L^1^** + (OBA^2−^)
	~220 (5.79/6.71)	240 − 800 (85.31/84.28)
**2**	MeOH + H_2_O ~220 (11.96/10.56)	**L^1^** + (SDA^2−^) 240 − 800 (80.07/81.29)
**3**	EtOH	**L^2^** + (OBA^2−^)
	~300 (7.29/4.52)	310 − 800 (85.86/84.77)
**4**	MeOH + H_2_O ~130 (5.11/5.62)	**L^2^** + (OBA^2−^) 290 − 800 (90.82/88.14)
**5**	8 H_2_O	**L^2^** + 2 (SDA^−2^)
	~180 (12.67/10.20)	270 − 800 (77.34/78.28)
**6**	2 EtOH + H_2_O ~170 (10.44/13.13)	**L^2^** + 2 (SDA^−2^) 200 − 800 (76.00/78.63)
**7**	4 H_2_O ~200 (10.09/9.97)	**L^3^** + (OBA^2−^) 260 − 800 (82.14/82.12)
**8**	4 H_2_O ~160 (9.40/9.71)	**L^3^** + (SDA^2−^) 230 − 800 (77.57/76.96)
**9**	0.5 EtOH + 2 H_2_O ~220 (10.08/9.26)	0.5 **L^3^** + (SDA^2−^) 240 − 900 (80.13/83.25)

**Table 3 ijms-23-03603-t003:** Average adsorbed iodine (mg g^−1^) for complexes **7**–**9** at different time intervals (min) and temperatures.

	7		8		9	
Time	25 °C	60 °C	25 °C	60 °C	25 °C	60 °C
30	0.208	0.379	0.258	0.606	22.73	107.92
60	0.379	0.549	0.606	1.298	63.72	140.64
120	0.521	0.587	0.952	1.558	118.93	161.30
180	0.587	0.606	1.298	1.558	159.75	165.15
360	0.606	0.606	1.385	1.558	163.14	166.55

**Table 4 ijms-23-03603-t004:** Average weight (%) from EDX for complexes **7**–**9**.

Element	7	8	9
C	56.93	55.70	49.45
O	23.54	22.04	25.66
Ni	11.34	11.07	9.68
I	0.42	0.01	7.03

**Table 5 ijms-23-03603-t005:** Crystallographic data for **1**–**9**.

**Compound**	**1**	**2**	**3**
Formula	C_28_H_26_NiN_4_O_9_	C_29_H_32_NiN_4_O_12_S	C_30_H_28_NiN_4_O_8_
Formula weight	621.24	719.35	631.27
Crystal system	Triclinic	Triclinic	Monoclinic
Space group	*P*ī	*P*ī	*P*2_1_/*c*
a, Å	10.2625(9)	12.0618(3)	10.4220(4)
b, Å	10.7071(9)	12.5407(3)	13.7045(5)
c, Å	13.8202(12)	12.6199(3)	19.3529(7)
α, °	70.272(5)	110.7299(12)	90
*β*, °	89.128(5)	99.6766(12)	91.645(2)
γ,°	81.387(5)	108.9309(12)	90
V, Å^3^	1412.3(2)	1595.49(7)	2763.00(18)
Z	2	2	4
D_calc_, Mg/m^3^	1.461	1.497	1.518
F(000)	644	748	1312
µ(Mo K_α_), mm^−1^	0.748	0.743	0.763
Range(2θ) for data collection,deg	1.566 ≤≤ 2θ ≤ 25.999	1.821 ≤ 2θ ≤ 28.308	1.821 ≤ 2θ ≤ 26.000
Independent reflections	5528 [R(Int) = 0.0234]	7907 [R(Int) = 0.0235]	5427 [R(Int) = 0.0300]
Data/restraints/parameters	5528/0/379	7907/1/442	5427/6/385
quality-of-fit indicator^c^	1.045	1.045	1.053
Final R indices [I > 2σ(I)] ^a,b^	R_1_ = 0.0384, wR_2_ = 0.0978	R_1_ = 0.0408, wR_2_ = 0.1196	R_1_ = 0.0407, wR_2_ = 0.1109
R indices (all data)	R_1_ = 0.0500, wR_2_ = 0.1039	R_1_ = 0.0510, wR_2_ = 0.1271	R_1_ = 0.0494, wR_2_ = 0.1163
**Compound**	**4**	**5**	**6**
Formula	C_29_H_26_NiN_4_O_8_	C_42_H_46_Ni_2_N_4_O_22_S_2_	C_46_H_50_Ni_2_N_4_O_20_S_2_
Formula weight	617.25	1140.37	1160.44
Crystal system	Monoclinic	Triclinic	Triclinic
Space group	*P*2_1_/*c*	*P*ī	*P*ī
a, Å	10.3888(3)	11.6188(10)	11.5864(5)
b, Å	13.6881(4)	12.3844(10)	12.3663(5)
c, Å	19.0647(7)	19.5572(16)	19.3944(8)
α, °	90	80.487(5)	82.136(2)
*β*, °	92.4267(17)	87.616(5)	87.810(2)
γ,°	90	63.355(4)	63.543(2)
V, Å^3^	2708.63(15)	2479.0(4)	2463.61(18)
Z	4	2	2
D_calc_, Mg/m^3^	1.514	1.528	1.564
F(000)	1280	1180	1204
µ(Mo K_α_), mm^−1^	0.777	0.929	0.743
Range(2θ) for data collection,deg	1.8332 ≤ 2θ ≤ 28.287	1.865 ≤ 2θ ≤ 28.434	1.821 ≤ 2θ ≤ 26.000
Independent reflections	6700 [R(Int) = 0.0197]	12315 [R(Int) = 0.0317]	12260 [R(Int) = 0.0216]
Data/restraints/parameters	6700/0/379	12315/0/651	12260/6/667
quality-of-fit indicator ^c^	1.047	1.030	1.048
Final R indices [I > 2σ(I)] ^a,b^	R_1_ = 0.0395, wR_2_ = 0.1102	R_1_ = 0.0448, wR_2_ = 0.1244	R_1_ = 0.0412, wR_2_ = 0.1239
R indices (all data)	R_1_ = 0.0440, wR_2_ = 0.1132	R_1_ = 0.0616, wR_2_ = 0.1345	R_1_ = 0.0455, wR_2_ = 0.1273
**Compound**	**7**	**8**	**9**
Formula	C_32_H_38_NiN_4_O_11_	C_32_H_38_NiN_4_O_12_S	C_24_H_26_NiN_2_O_9_._5_S
Formula weight	713.37	761.43	585.24
Crystal system	Monoclinic	Monoclinic	Triclinic
Space group	*C*2/*c*	*C*2/*c*	*P*ī
a, Å	20.2189(10)	19.5355(6)	7.1192(2)
b, Å	9.4113(5)	9.5220(3)	11.0064(3)
c, Å	18.7496(9)	19.2870(5)	17.0108(5)
α, °	90	90	89.7201(17)
*β*, °	102.6301(18)	103.1949(11)	78.8806(18)
γ,°	90	90	81.4570(19)
V, Å^3^	3481.5(3)	3492.99(18)	1292.99(6)
Z	4	4	2
D_calc_, Mg/m^3^	1.361	1.448	1.503
F(000)	1496	1592	608
µ(Mo K_α_), mm^−1^	0.620	0.683	0.888
Range(2θ) for data collection,deg	2.226 ≤ 2θ ≤ 25.998	2.169 ≤ 2θ ≤ 28.286	1.872 ≤ 2θ ≤ 28.356
Independent reflections	3424 [R(Int) = 0.0465]	4322 [R(Int) = 0.0292]	6421 [R(Int) = 0.0431]
Data/restraints/parameters	3424/0/236	4322/1/236	6421/3/353
quality-of-fit indicator ^c^	1.036	1.069	1.039
Final R indices [I > 2σ(I)] ^a,b^	R_1_ = 0.0532, wR_2_ = 0.1488	R_1_ = 0.0371, wR_2_ = 0.0976	R_1_ = 0.0440, wR_2_ = 0.0977
R indices (all data)	R_1_ = 0.0646, wR_2_ = 0.1645	R_1_ = 0.0439, wR_2_ = 0.1038	R_1_ = 0.0760, wR_2_ = 0.1089

^a^ R_1_ = ∑||F_o_| − |F_c_||/∑|F_o_|. ^b^ wR_2_ = [∑w(F_o_^2^ − F_c_^2^)^2^/∑w(F_o_^2^)^2^]^1/2^. w = 1/[σ^2^(F_o_^2^) + (ap)^2^ + (bp)], p = [max(F_o_^2^ or 0) + 2(F_c_^2^)]/3. a = 0.0511, b = 0.0836 for **1.** a = 0.0725, b = 0.7345 for **2**. a = 0.0407, b = 0.0494 for **3**; a = 0.0395, b = 0.0440 for **4.** a = 0.0448, b = 0.0616 for **5**. a = 0.0412, b = 0.0455 for **6**; a = 0.0532, b = 0.0646 for **7.** a = 0.0371, b = 0.0439 for **8**. a = 0.0440, b = 0.0760 for **9**. ^c^ quality-of-fit = [∑w(|F_o_^2^| − |F_c_^2^|)^2^/N_observed_ − N_parameters_)]^1/2^.

## Data Availability

Data is contained within the article or Appendix A.

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
