# Peer review of "Nickel(II) Coordination Polymers Supported by Bis-pyridyl-bis-amide and Angular Dicarboxylate Ligands: Role of Ligand Flexibility in Iodine Adsorption"

_ijms, 2022, doi:10.3390/ijms23073603_

Round 1

Reviewer 1 Report

The authors describe the synthesis of 9 Ni(II) complexes containing bis-pyridyl-bis-amide and angular dicarboxylato ligands forming coordination polymers. These compounds have been used for I2 adsorption experiments. The complexes were characterized by use of X-ray diffraction methods as well as elemental analyses. In my opinion the novelty of that contribution is rather low showing routine work. The description of the the 9 molecular structures seems to me tiresome. For the reviewer it is not accessible why exactly these CPs are used for iodine adsorption and what the benefit is. For these reasons I regret not being able to recommend that manuscript for publication in IJMS.

Reviewer 2 Report

The authors describe an article entitled “Nickel(II) Coordination Polymers Supported by Bis-pyridyl-bis amide and Angular Dicarboxylate Ligands: Role of Ligand Flexibility in Iodine Adsorption”. The topic of the manuscript is interesting, and the manuscript constitutes an interesting research article concerning coordination polymers.

The work is well-written and a well-constructed introduction has been established by the authors. Sufficient spectra and figures are included in the manuscript for comprehension and clarity. Interesting and convincing results are also presented in this work. Overall, I think that this is a manuscript that I recommend for publication after inclusion of minor revisions.

1) NMR analyses of complexes should be added.

2) What about the absorption spectra of the different complexes. It should be added.

3) Electrochemical properties of the different complexes should be investigated.

4) what about the thermal stability of the different complexes. TGA and DSC measurements should be determined.

For all the above-mentioned reasons, at present, do not publish.   

Round 2

Reviewer 1 Report

The authors has explained in their response to my review report why this contribution could deserve for publication in IJMS. Undoubtedly, the experiments are carefully carried out, but considering the high impact of this journal, in my opinion the novelty of that work is too low. Nevertheless, I do not want to prevent a publication in that journal, although I am not really convinced.